# Building safer and more resilient cities in China: A novel approach using a dynamic nonhomogeneous Gray model for data-driven decision-making

Jian Liu[1,2], Ye He[1], Rui Feng ☉[1,3]*, Bin Lyu[1]

1 School of Civil and Resource Engineering, University of Science and Technology Beijing, Beijing, People's Republic of China, 2 Key Laboratory of High-Efficient Mining and Safety of Metal Mines of the Ministry of Education, University of Science and Technology Beijing, Beijing, People's Republic of China, 3 Research Institute of Macro-Safety Science, University of Science and Technology Beijing, Beijing, People's Republic of China

* fengr@ustb.edu.cn

**Data Availability Statement:** All relevant data are within the manuscript and its Supporting Information files.

## Abstract

Urban resilience is crucial for sustainable development and resident safety in a changing environment with potential risks. Given China's rapid urbanization, constructing resilient cities that anticipate risks, mitigate disaster impacts, and swiftly recover from crises is paramount. This study explores a key area of urban construction: building safety. We apply the dynamic nonhomogeneous grey model (DNMGM(1,1)) to simulate the building death toll and use a traffic accident death toll dataset for validation. Unlike traditional models, DNMGM(1,1) can integrate and respond to new data points in real-time, thus producing accurate predictions when facing new trends or fluctuations in the data. The research findings indicate that with a dataset size of 6, the DNMGM(1,1) model achieves average relative errors of 9.26% and 7.29% when predicting fatalities in both construction and traffic accidents. This performance demonstrates superior prediction accuracy compared to traditional grey models. This method uses prediction models to support the construction of elastic cities, providing strong data support and decision-making tools for planning and resource allocation. Specific interventions and policy frameworks based on this study by urban planners and policymakers can promote resilient urban development. Future efforts should strive to enhance its robustness and adaptability in different fields.

## 1 Introduction

In the midst of the swift progress of global urbanization, the construction industry assumes a pivotal role across numerous nations, particularly in rapidly developing countries like China. Serving as a cornerstone of the national economy, the Chinese construction sector witnessed significant expansion in the year 2022, attaining a comprehensive output value of 31.2 trillion yuan, constituting 25.29% of China's Gross Domestic Product. In comparison to the preceding year, the overall output value of the construction industry saw a notable uptick of 6.48%, concurrently generating substantial employment opportunities for a workforce totaling 51.84

**Funding:** This study was funded by grants from the National Natural Science Foundation of China (52004139), the Key Technologies Research and Development Program (2017YFC0804901), and Fundamental Research Funds for the Central Universities (No. FRF-TP-22-120A1).

**Competing interests:** The authors have declared that no competing interests exist.

million individuals [1]. Nevertheless, amid the continuous evolution of China's construction sector, there is a discernible rise in societal apprehension concerning issues related to construction safety. Pertinent statistical data highlights that the economic ramifications stemming from safety incidents in construction have escalated to the extent of ten percent of the aggregate costs of construction projects. This not only acts as an impediment to the sustainable progress of the construction industry, leading to direct financial losses, but also exerts an adverse impact on social cohesion and stability [2].

In the expansive landscape of China's construction industry, marked by intricate construction processes and a multitude of influencing factors, safety incidents not only interrupt routine production activities but also present substantial risks to the well-being of construction workers and result in considerable property damage. For a society striving for high-quality and sustainable development, effectively reducing the incidence and mortality rates of such accidents is an urgent issue. Against this backdrop, the concept of resilient cities has emerged [3]. The concept of resilient cities extends beyond physical infrastructure to encompass the soft power of urban areas, namely, how to maintain city functions and services in the face of various risks and challenges, ensuring residents' safety and quality of life [4]. Therefore, the ability to anticipate risks and thereby strengthen accident prevention is particularly crucial. For instance, predicting the number of fatalities in construction can aid city managers in identifying potential danger zones and taking measures to mitigate accidents and disasters. By accurately forecasting the likelihood of construction accidents, city managers can proactively implement preventative measures such as enhancing on-site safety management, improving construction techniques, and bolstering workers' safety awareness and training, thereby effectively avoiding or mitigating accidents. In addition, through other angles, such as strengthening safety law enforcement, safety supervision, etc., can also effectively prevent the occurrence of accidents [5]. The key to truly enhancing the resilience of resilient cities to withstand emergencies lies in establishing and implementing an efficient accident prediction and prevention mechanism and introducing a more accurate fatality prediction model that predicts the durability of buildings under extreme conditions. The practical implementation of this model to optimize safety through data-driven decision-making has the potential real-world impact of significantly improving the safety and security of residents and guiding the transformation and upgrading of the construction industry to promote safer and more sustainable urban development.

Currently, a variety of prediction methods have been developed and utilized across numerous fields, including neural networks [6–13], ARIMA models [14–17], regression models [18–22], and support vector regression [23–28], among others. Among these predictive technologies, the grey model (GM), with its distinctive "small data" characteristic, has been widely applied in forecasting across various industries [29], energy [30–32], construction [33], and environmental sectors [34–36]. Deng [37] introduced the grey system theory, specifically designed for systems dealing with incomplete and uncertain information. G.M. (1,1) [38] and DGM(1,1) (Discrete Grey Models) [39–41] are the most prevalent univariate grey prediction models, primarily utilized for modeling and predicting approximately homogeneous exponential growth sequences. They boast advantages such as a straightforward modeling process, minimal data requirements, and ease of use.

However, construction safety events, characterized by complexity and uncertainty, often manifest as approximately non-homogeneous sequences. In such contexts, achieving satisfactory simulation and prediction accuracy can be challenging due to the structural constraints of the G.M. (1,1) or DGM(1,1) models. Fortunately, substantial research progress has been made in grey modeling for non-homogeneous exponential sequences, which can be categorized into three groups based on different modeling approaches: (1) Tong [42] introduced a whitening

differential equation to construct approximate non-homogeneous grey prediction models. By solving this equation, the one-time accumulation sequence of the original sequence is derived, and the recursion equation of this sequence, combined with Cramer's rule, is used to solve the system of equations and obtain the coefficients. This method, known as the DNGM(1,1) model (Discrete Non-homogeneous Grey Model), directly derives parameter solutions from the differential equations, offering superior predictive performance compared to traditional methods that estimate parameters through differential equations, thereby providing a more robust prediction framework. (2) In sequence modeling, transforming an approximately non-homogeneous exponential sequence into an approximately homogeneous one is crucial to meet the homogeneity requirements of DGM(1,1) or G.M. (1,1) models. To achieve this, Zeng [43] employed the concept of cumulative generation in modeling time series data. The uniqueness of this model lies in its foundation on the cumulative generation sequence of the original data, rather than directly modeling the original sequence. This transformation led to the creation of an indirect DGM(1,1) model, known as the IDGM(1,1) model (Improved Discrete Grey Models). (3) If the initial sequence already exhibits characteristics of approximately non-homogeneous exponential growth, applying cumulative generation based on the exponential law may disrupt the existing pattern, potentially hindering the desired cumulative generation effect. In such scenarios, utilizing a direct grey model for modeling the original sequence becomes a viable approach. Wang [44] proposed a direct modeling method for G.M. (1,1), which models the raw data directly without the need for accumulation and has demonstrated favorable results in specific data scenarios. Inspired by this, Zeng and Liu [45] introduced a similar method, bypassing the cumulative generation steps, resulting in the DDGM(1,1) model (Development and Discrete Grey Models).

Nevertheless, despite their advantages, the linear characteristics of the classical G.M. (1,1), DGM(1,1), and even the improved DNGM(1,1), IDGM(1,1), and DDGM(1,1) models may still lead to inaccurate predictions in specific practical contexts. Researchers are constantly exploring new prediction methods, such as SEM (structural equation model) [46], which combines the characteristics of factor analysis and path analysis, and R.F. algorithm (Random Forest) [47], which can simultaneously process multiple observed variables and potential variables, and construct multiple decision trees and summarize their prediction results to improve prediction accuracy and stability. However, they usually have the disadvantage of high computational cost.

This article delves into the constraints inherent in the linear G.M. (1,1) model and introduces an innovative approximate non-homogeneous dynamic grey model, namely the DNMGM(1,1) model, with the objective of enhancing the precision in predicting fatalities resulting from construction accidents in China. The proposed model integrates the DNGM model, starting from a whitening differential equation. This strategic decision ameliorates errors stemming from employing difference equations for parameter estimation, enhancing its suitability for modeling non-homogeneous exponential sequences. As prediction accuracy in grey models is intricately linked with the data volume, this study integrates optimal size operations into the dynamic model. This operation reflects the new information priority principle, thereby enhancing the adaptability of the model.

## 2 Approximate inhomogeneous dynamic GM(1,1) modeling process

### 2.1 Traditional modeling process and its errors

In the traditional modeling process, parameters A and B are obtained through the least squares estimation method to simplify the difference equation. The values of A and B in the final

restoration equation are derived by solving the whitening equation, resulting in errors in the model.

So starting from the whitening equation, get Eq (1):

$$\frac{dx^{(1)}}{dt} + Ax^{(1)} = tB + C \tag{1}$$

The reconstructed whitening equation corresponding to the homogeneous equation is:

$$\frac{dx^{(1)}(t)}{dt} + Ax^{(1)}(t) = 0 \tag{2}$$

The common solution can be obtained:

$$x^{(1)}(t) = C_1 e^{-At} \tag{3}$$

Based on the constant change method, in Eq (3), $C_1$ can be replaced with $m(t)$ to obtain:

$$x^{(1)}(t) = m(t)e^{-At} \tag{4}$$

Taking the derivative of Eq (4) on both sides results in:

$$\frac{dx^{(1)}(t)}{dt} = m'(t)e^{-At} - Am(t)e^{-At} \tag{5}$$

Moreover, substituting into Eq (1) yields:

$$m'(t)e^{-At} - Am(t)e^{-At} = Bt + C - Ax^{(1)}(t) \tag{6}$$

Substituting Eqs (4) into (6) produces:

$$m'(t) = (Bt + C)e^{At} \tag{7}$$

By integrating both sides of Eq (7), we obtain:

$$m(t) = \int (Bt + C)e^{At}dt == \frac{B}{A}te^{At} - \frac{B}{A^2}e^{At} + \frac{C}{A}e^{At} + C_2 \tag{8}$$

Substituting Eqs (8) into (4) yields:

$$x^{(1)}(t) = \left(\frac{B}{A}te^{At} - \frac{B}{A^2}e^{At} + \frac{B}{A}e^{At} + C_2\right)e^{-At} = \frac{B}{A}t - \frac{B}{A^2} + \frac{C}{A} + C_2 e^{-At} \tag{9}$$

When $t = 1$, $x^{(1)}(1) = C_2 e^{-A} + \frac{B}{A} - \frac{B}{A^2} + \frac{C}{A}$, the following expression is derived:

$$C_2 = \frac{x^{(1)}(1) - \frac{B}{A} + \frac{B}{A^2} - \frac{C}{A}}{e^{-A}} \tag{10}$$

Thus, get the time response expression:

$$x^{(1)}(t) = \left(x^{(1)}(1) - \frac{B}{A} + \frac{B}{A^2} - \frac{C}{A}\right)e^{-A(t-1)} + \frac{B}{A}t - \frac{B}{A^2} + \frac{C}{A} \tag{11}$$

The final reduction formula is:

$$\hat{x}^{(0)}(t) = \hat{x}^{(1)}(t) - \hat{x}^{(1)}(t-1) = (1 - e^A)\left(x^{(1)}(1) - \frac{B}{A} + \frac{B}{A^2} - \frac{C}{A}\right)e^{-A(t-1)} + \frac{B}{A} \tag{12}$$

Utilizing the homogeneous equation derived from Eq (11), we can obtain the recursive equation system as:

$$x^{(1)}(t+1) = e^{-A}x^{(1)}(t) + \frac{B}{A}(1 - e^{-A})t + (1 - e^{-A})\left(\frac{C}{A} - \frac{B}{A^2}\right) + \frac{B}{A} \tag{13}$$

Let:

$$\alpha = e^{-A}, \beta = \frac{B}{A}(1 - e^{-A}), \gamma = (1 - e^{-A})\left(\frac{C}{A} - \frac{B}{A^2}\right) + \frac{B}{A} \tag{14}$$

Thus, we obtain:

$$x^{(1)}(t+1) = \alpha x^{(1)}(t) + \beta t + \gamma \tag{15}$$

Solve this recurrence equation to obtain simulated and predicted values.

## 2.2 The necessity and introduction process of dynamic modeling

Factors affecting forecasts change with system state. When new modeling data is continuous, the original data is unnecessary. Extract new data and replace the oldest based on a specified size. This approach fully uses the updated information and the idea of "metabolism" [48] to build dynamic models. Add new data after a sequence, delete old known data, keep the same dimensions of the data sequence, and constantly supplement new information through such metabolic thinking.

Based on the above ideas, construct the structural diagrams of four models as shown in Fig 1. Fig 1 shows a complete process from data preprocessing to model building, parameter estimation, optimal data size determination, and dynamic modeling to improve the accuracy and robustness of time series predictions. A "metabolic process" step has been added for dynamic models to update the model's metabolism. This dynamic process can be used to update the prediction system effectively and avoid the loss of accuracy in static processes. Choosing the classic G.M. (1,1) or DNMGM(1,1) is an important step in dynamic modeling, and this step is marked in green. Different data sizes may lead to different prediction results and different prediction efficiencies. The static model only uses a random dataset size, and the effect of the dataset size on accuracy is not considered. Therefore, the final proposed model is established after determining the optimal dataset size, and this dynamic process is marked in blue.

## 2.3 Comparison of characteristics of four models

Analyzing the structural diagrams of the four models for static and dynamic processes reveals that G.M. (1,1) originates directly from the original data sequence, providing predictions based on cumulative generation and grey differential equations. This model is relatively simple and does not involve any external factors or dynamic adjustments. DNGM(1,1) introduces parameter estimation for the whitening differential equations, transforming the model from a homogeneous equation to a nonhomogeneous equation and enhancing the consideration of nonlinear changes in the data. MGM(1,1), building upon the foundation of G.M. (1,1), introduces the concept of "metabolism," allowing the model to transform or adjust data to better capture trends. The introduction of the metabolic concept increases the complexity of the model but also enhances its predictive capabilities. Moreover, DNMGM(1,1) merges DNGM (1,1) and MGM(1,1) traits. It estimates parameters in whitening differential equations and uses metabolism. The most complex model is also the most capable of handling intricate data patterns.

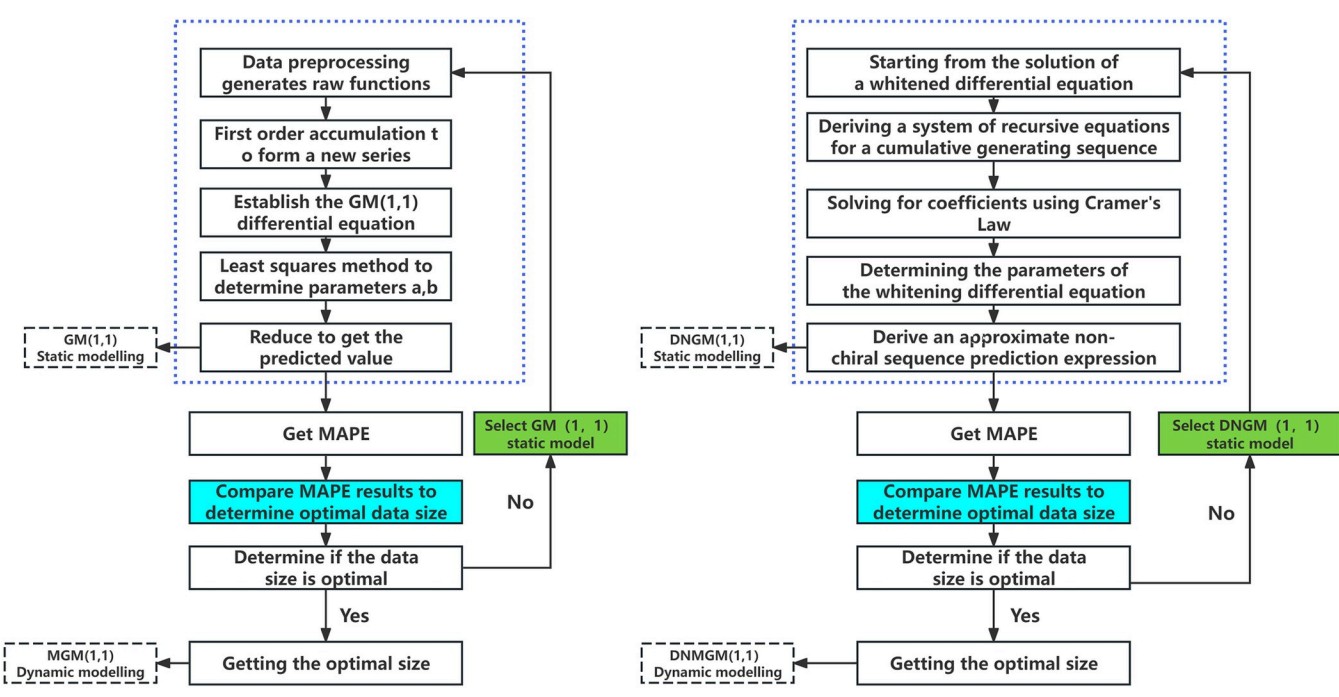

**Fig 1. Flowchart of data preprocessing and model selection for grey prediction models.**

## 2.4 Application of DNMGM(1,1) model

The classic G.M. model uses a random dataset size, disregarding optimal size for accuracy. Identifying the optimal size for simulated data reveals data trends. Different data amounts yield varied predictions and efficiency. Therefore, determining the optimal dataset size (n) is important for improving the accuracy of predictions. This study introduces the DNMGM(1,1) model, an approximate non-homogeneous dynamic grey model. It is developed on the foundation of the DNGM(1,1) model, incorporating it with the MGM(1,1) model, and is applied to predict fatalities in construction accidents. In the construction industry, the practical implementation of the DNMGM(1,1) model is of great value. For example, we can imagine integrating this model into existing building safety protocols, through specific implementation steps or developing specialized tools, to analyze and predict the risk of building accidents in real-time, thereby providing scientific guidance to governments to help them adjust safety investments and optimize the allocation of safety personnel. Such applications can improve the overall safety of the construction industry and significantly reduce accident rates and fatalities. Construction accidents occur less frequently than other data categories (such as mine accidents), which means the data collected may be sparse, making the prediction task more difficult. At the same time, construction fatalities may be influenced by various factors, such as regulatory changes, the introduction of new technologies, or economic fluctuations. These factors make the data exhibit nonlinearity and instability. Through the grey model, we can make effective predictions when the data is sparse.

In order to further demonstrate the advantages of the DNMGM(1,1) model, we conducted a hypothetical or practical comparative analysis. Compared with other prediction models (such as traditional time series analysis models, machine learning models, etc.), the DNMGM (1,1) model captures the nonlinearity and instability characteristics of the data more accurately through its unique dynamic method and iterative calculation process, thus achieving a

significant improvement in prediction accuracy. These comparative results not only enhance the theoretical value of this study, but also provide strong support for its promotion in practical applications.

## 3 Forecasting of the death risk in China

There are various indicators of construction accidents, including fatalities, accident rates, and fatalities per 100,000 people. This study conducts a statistical analysis of construction accident fatality data in China from 2000 to 2019 [49]. As shown in Fig 2, he number of construction-related deaths in China showed an upward trend from 2000 to 2003. Subsequently, there was a continuous decline from 2003 to 2015, indicating an improvement in building safety conditions. However, deaths have increased again since 2015, highlighting ongoing safety and security challenges. The contradiction between building development and building safety is still prominent.

The death toll directly reflects the severity of the accident and the direct impact on human life, which is one of the most direct and significant consequences of the accident. It is not only an important factor in classifying the accident level, but it also helps to classify accident management so that the relevant departments can quickly take corresponding emergency measures and follow-up treatment according to the accident level. At present, safety guidance is still facing challenges, especially in curbing major accidents and reducing a large number of deaths. Accurately predicting the development trend of building safety is beneficial for transforming post emergency measures into pre prevention measures, and can effectively promote the

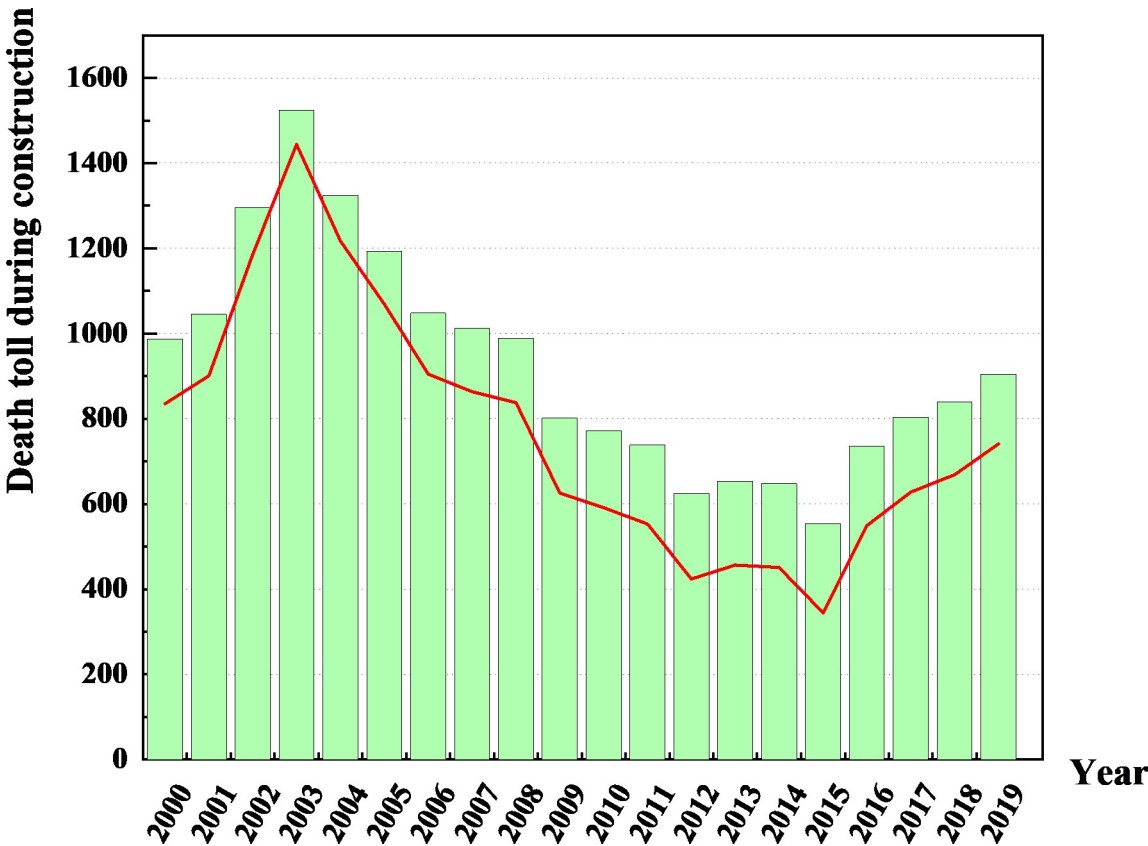

**Fig 2. Trend of building construction fatalities in China from 2000 to 2019.**

efficient implementation of current building safety work. Construction accident data are characterized by incomplete information and uncertainty. Therefore, the grey prediction model is used for prediction.

## 3.1 Static prediction results and analysis

The data used in this paper are from the official statistical yearbook of the Ministry of Housing and Urban-Rural Development of China. The statistical yearbook is regularly compiled and released by authoritative institutions, aiming to comprehensively and systematically reflect the development status, changing trends, and existing problems in the field of housing and urban and rural construction in China. Our data is carefully screened and collated to ensure its accuracy, authority, and timeliness. These data not only provide a solid foundation for the research of this paper, but also further enhance the credibility and persuasiveness of the research conclusions.

Utilizing the G.M. (1,1), DGM(1,1), DNGM(1,1), and IDGM(1,1) models, we established five models to forecast the number of fatalities resulting from construction accidents in China spanning from 2000 to 2019. The results are shown in Fig 3, which visually presents the comparison of the data predicted by different models on the number of deaths, while providing a reference base for real data. The solid black line represents the actual number of deaths, and the different colored lines represent the predicted values of the various models. IDGM model (red dashed line) and DDGM model (orange dashed line) are quite different from the reality. It is worth noting that the remaining three dashed lines differ less from reality, but the DNGM model represented by the green dashed line is closest to reality at some extreme points

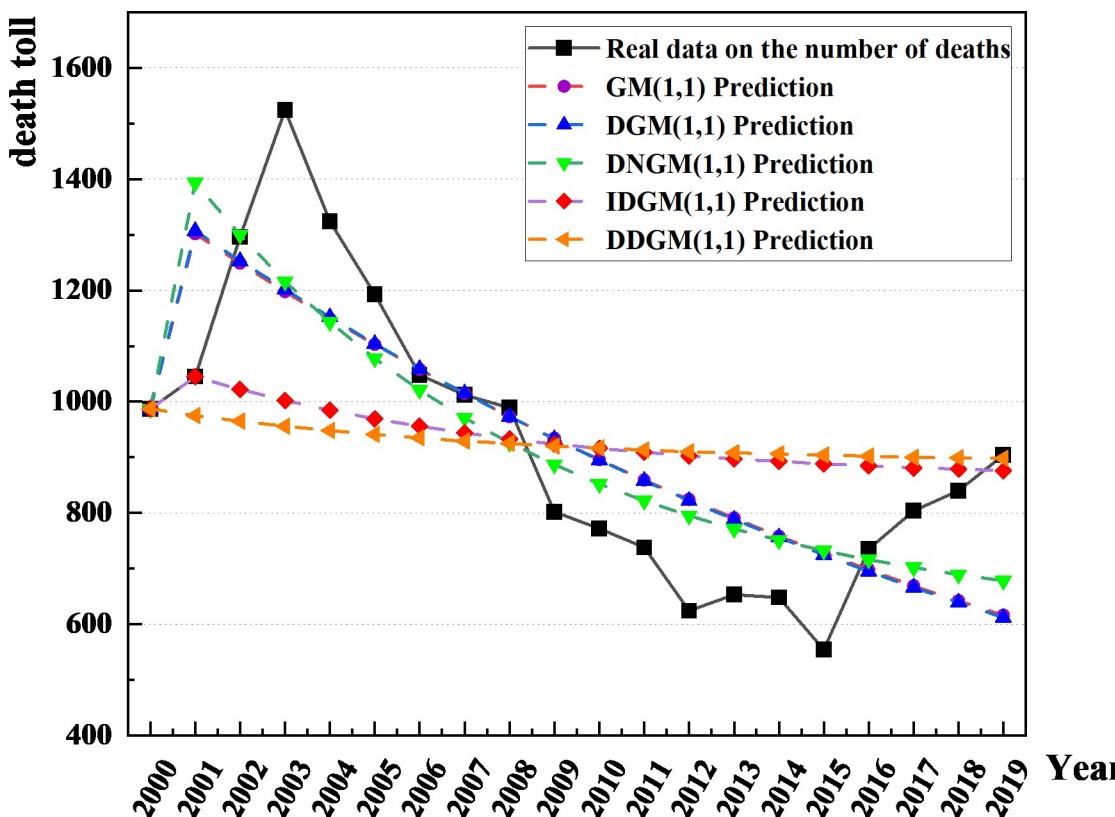

**Fig 3. Five static models for predicting the number of deaths in construction accidents versus the actual number.**

compared to the DGM model (purple dashed line) and the DGM model (blue dashed line), indicating that the DNGM model shows greater accuracy in predicting the actual number of deaths.

To rigorously and judiciously assess the outcomes of model simulations, the adoption of appropriate evaluation methods becomes imperative. This study employs the MAPE to judge the model's suitability. In this study, MAPE (Mean Absolute Percentage Error) was selected as the core index to quantify and evaluate the applicability of each model in predicting the number of deaths from construction accidents in China. As an assessment indicator, MAPE helps to quantify the model prediction accuracy, visually reflects the degree of prediction deviation from the true value through the form of a percentage, and has a guiding role in improving the model prediction performance. As shown in Fig 4, the relative error performance of the five different models in the prediction process emerged, which provides a basis for in-depth analysis. Upon scrutiny of the forecast results, the G.M. (1,1) model and the DGM(1,1) model exhibit average relative errors of 15.03% and 15.01%, respectively, in predicting construction-related deaths. The observed average relative error falls within the range of 10–20%. However, a striking phenomenon is that the relative error values of both models have shown a continuous upward trend since 2009, which is likely to be closely related to non-stationary fluctuations in accident deaths in recent years, especially when the data series lacks strict monotonically increasing characteristics. Traditional exponential growth models such as G.M. (1,1) and DGM(1,1) are inadequate to capture this complex dynamic change.

In contrast, DNGM(1,1), IDGM(1,1), and DDGM(1,1) models show different performance in prediction accuracy. The DNGM(1,1) model stands out with a MAPE value of 13.73%, which is not only better than the DGM(1,1) model but also its error control is in the ideal range of 10% to 20%, which indicates that the model has advantages in adapting to the complex changing pattern of the death toll of construction safety accidents in China. On the

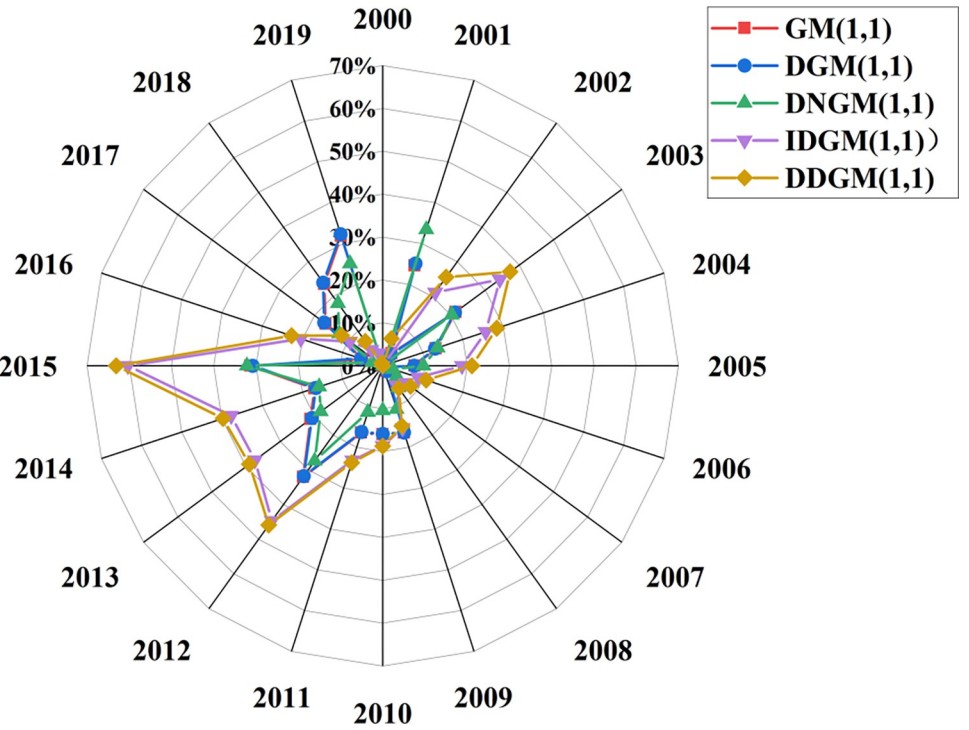

**Fig 4. The precision values of the simulation results of the five models.**

contrary, the MAPE values of IDGM(1,1) and DDGM(1,1) models were 19.79% and 21.59%, respectively, which were significantly higher than those of DNGM(1,1) models, reflecting their deficiency in prediction accuracy. This difference deeply reveals the differences in sensitivity and adaptability of different model structures to specific data features, especially when faced with a data set with a unique downward trend, such as the number of fatalities from construction safety accidents in China.

Further, through comprehensive analysis of the relative error values and their changing trends of the five models, it can be reasonably inferred that under the framework of static models, the DNGM(1,1) model becomes the optimal model for predicting the risk related to building death in China due to its low prediction error and good adaptability. This divergence can be ascribed to the distinctive features of construction safety accidents in China, where the overall trend manifests a decreasing trajectory gradually stabilizing over time. Consequently, through a comprehensive analysis of the relative error values among the five models, it can be deduced that, within the realm of static models, the DNGM(1,1) model stands as a more suitable choice for forecasting the risks associated with construction fatalities in China.

### 3.2 Dynamic prediction results and analysis

Based on an analysis of the static model for construction fatalities in our country, a novel grey model, the DNMGM model, has been developed. This model integrates the approximate nonhomogeneous DNGM(1,1) model with the dynamic MGM model. Leveraging the dynamic features of the MGM model facilitates the transformation of the nonlinear grey model from a static to a dynamic framework. This enables real-time integration and responsiveness to emerging data points, effectively revealing and tracking new trends or fluctuations in the data.

Static models tend to lose more predictive information regarding processes, whereas dynamic models can effectively harness new information. The optimal value of n in dynamic models needs iterative calculation and comparison, and the choice of n can significantly impact predictive performance in most simulation scenarios. Adjusting the appropriate dataset size can enhance prediction accuracy. In general, a limited amount of data may be insufficient for constructing predictive models, while an abundance of data runs the risk of degrading into static models.

This study selected various values of n, namely 4, 5, 6, 7, 8, 9, and 10, to obtain different average relative errors, as illustrated in Fig 5. The average relative error served as the evaluation metric. MAPE represents prediction errors in percentage terms, allowing us to directly compare prediction performance between different size datasets or different models, as percentage errors eliminate the effect of the magnitude of the data on the evaluation results. At the same time, MAPE is more sensitive to prediction error. Even a small absolute error will result in a high percentage error when the actual value is small. MAPE is obtained by calculating the percentage difference between the predicted and actual values at each point in time and then taking the average of the absolute values of these percentage errors. This evaluation index directly reflects the accuracy of the forecast. By comparing MAPE at different n values, we chose the n value that produces the smallest MAPE (i.e., 6) as the optimal dataset size. When the dataset size is 6, the MAPE of Chinese building death data reaches the minimum value, and the MAPE generated in the prediction process is less than 10%, which indicates that the model has high prediction accuracy under the dataset size.

Dynamic models, particularly MGM (1,1) and DNMGM (1,1) models, were utilized with a data size of 6 for forecasting construction accident fatalities in China from 2000 to 2019. Comparative analyses were conducted to assess the predicted values of G.M., DNGM, MGM, and DNMGM models against the actual values, as depicted in Fig 6. Furthermore, the results of

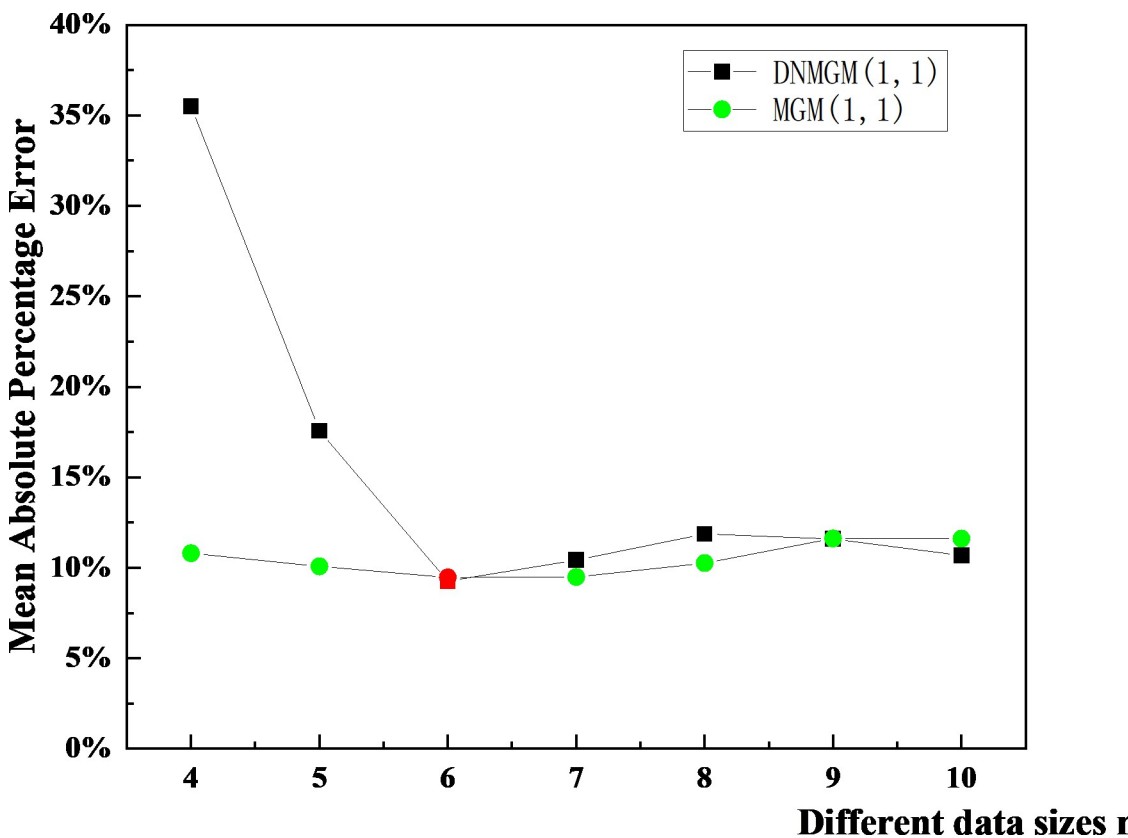

**Fig 5. The optimal dataset size for construction death prediction in China with the MGM and DNMGM models.**

annual relative error predictions are illustrated in Fig 7, highlighting the superior prediction accuracy of the DNMGM model. The average relative errors for G.M., DNGM, MGM and DNMGM models in forecasting fatalities from construction safety accidents are 15.03%, 13.73%, 9.46%, and 9.26%, respectively. The MAPE exhibits a gradual decline, indicative of continuous improvement in model outcomes. Fig 6 demonstrates that static models consistently lag behind the actual curve towards the later stages. This disparity arises from the inherent limitation of static models relying on fixed data sizes for predicting longer-term future data. Consequently, as the years increase, the greater the time distance between the model's simulated data and future forecast data, the greater the error, and the greater the deviation between the curve and the actual data curve. In contrast to static models, dynamic models, through real-time adjustment of internal parameters, can adeptly capture the current state and potential trends of the data. This dynamic adjustment mechanism empowers the model to sustain high prediction accuracy amid rapid changes in data, ensuring that the annual relative error remains concentrated within the 20% threshold.

## 4 Discussion

### 4.1 The dynamic adaptive capability and generalizability of the DNMGM (1,1) model

By using the DNMGM(1,1) model to predict construction fatalities in risk identification and assessment, city managers can identify potentially high-risk building areas or construction phases, and the predictions can be combined with other city data (e.g., traffic flow and

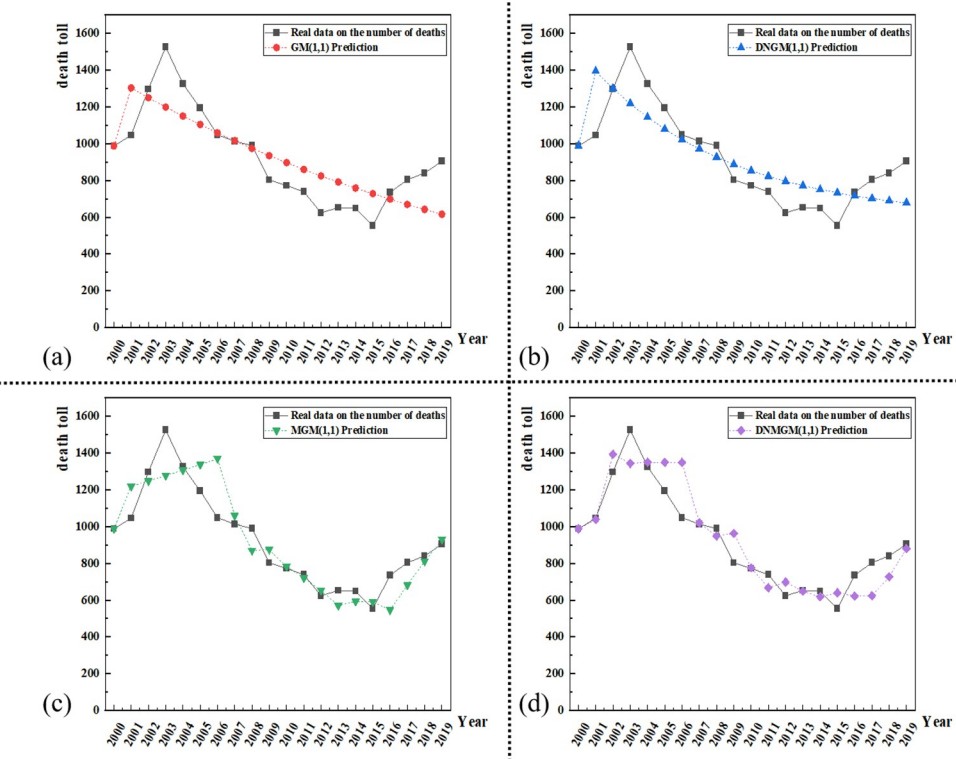

**Fig 6. Comparison of the true and predicted values of the number of fatalities from construction safety accidents predicted by the four models.**

population density). To further identify and assess points of risk in a city, in resource allocation and planning, based on the predicted number of construction fatalities, cities can prioritize the allocation of resources to areas that are projected to be at high risk, such as increasing supervision and inspections and providing additional safety training. These measures can help ensure that the buildings and infrastructure can withstand changes in the future, thus enhancing the city's resilience. Furthermore, if the model predicts a potential increase in construction accident fatalities in a specific region or during a particular period, an early warning system can proactively issue alerts to notify relevant authorities. In policy formulation and adjustment, based on model predictions, the government and urban planners can formulate or adjust relevant construction and safety policies to ensure the safety and durability of urban buildings. For example, if the risk of construction fatalities in a certain area is predicted to be high, more stringent building standards or higher safety management requirements for construction sites can be considered; at the same time, publicizing the predictions of models can improve the public's confidence in a city's ability to cope with various challenges. Additionally, this approach can raise the public's awareness of risk and encourage them to participate in the construction of resilient cities. As a result, through accurate prediction and analysis, cities can better cope with challenges and ensure the safety and well-being of their inhabitants.

The application of the DNMGM(1,1) model in construction accidents provides a powerful basis for analyzing and predicting construction safety. However, the concept of a resilient city is not limited to construction safety and is multidimensional, including social and economic systems, transportation systems, environmental protection, and other factors. To investigate the universality of the DNMGM(1,1) model, we selected three grey prediction models, namely,

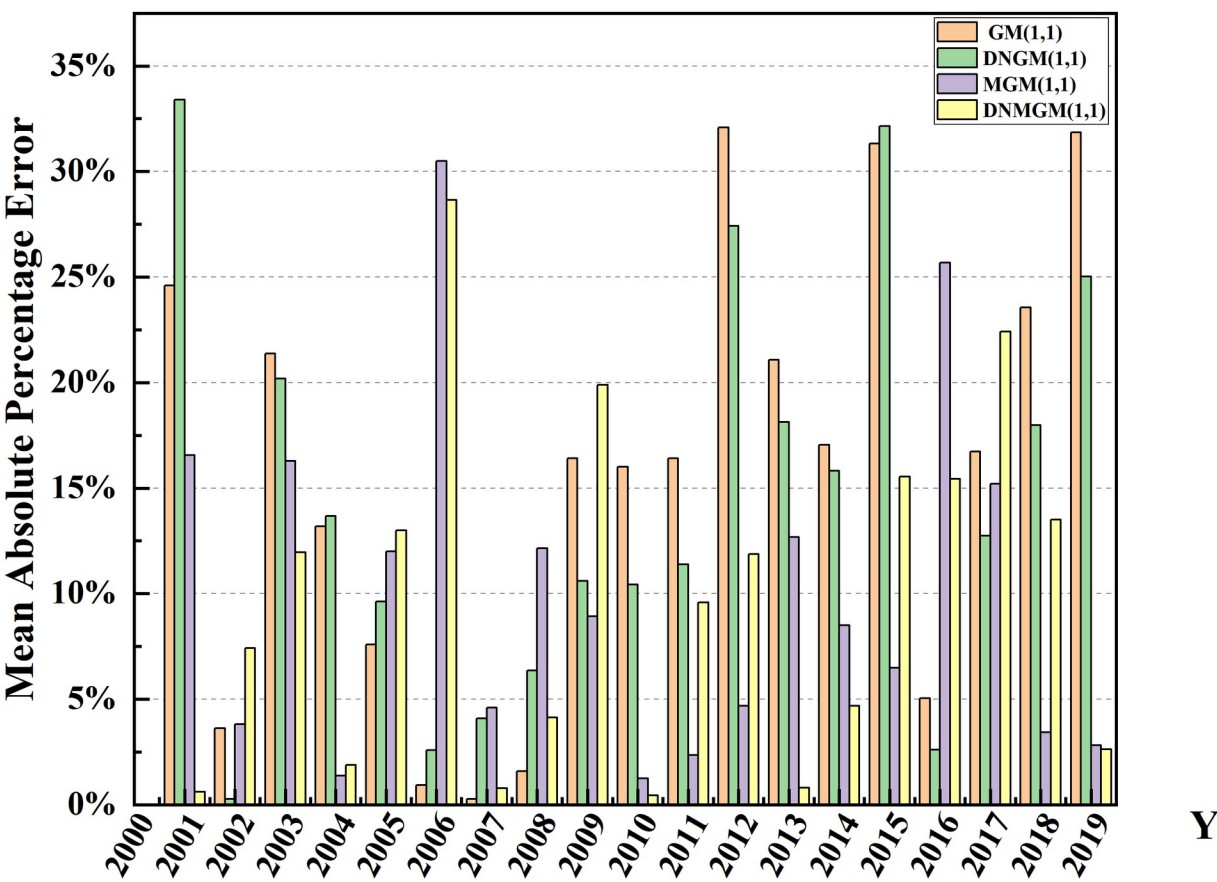

**Fig 7. MAPE of the four models for predicting construction safety fatalities from 2000 to 2019.**

G.M. (1,1), DNGM(1,1), and DNMGM(1,1), to predict the number of accident-related deaths in the traffic field from 2010 to 2019.

Various dataset sizes, ranging from 4 to 10, were considered, and the DNGM(1,1) model was applied to predict accident fatalities in the transportation sector from 2010 to 2019. The dynamic DNMGM(1,1) model results are presented in Table 1, demonstrating superior prediction accuracy at a forecast size of 5 or 6, with average relative errors of 7.36% and 7.29%, respectively. Optimal sizes for modeling the dynamic DNMGM(1,1) were determined based on lower MAPE values, resulting in the DNMGM(1,1)(n = 5) and DNMGM(1,1)(n = 6) models. Employing the G.M. (1,1), DNGM(1,1), DNMGM(1,1)(n = 5), and DNMGM(1,1)(n = 6) models for predicting transportation sector accident fatalities from 2010 to 2019, Table 2 reveals that dynamic models, particularly those with a forecast size of 5 or 6, exhibit substantially lower average relative errors compared to the static G.M. (1,1) model (17.35%) and DNGM(1,1) model (13.26%). Fig 8 illustrates a comparison of predicted values from the four models with actual values. Notably, the trends align more closely with the actual trajectory of accident fatalities in the transportation sector, affirming the continued applicability of the

**Table 1. MAPE of simulation results for DNMGM(1,1) with different dataset sizes (4–10).**

| Different dataset sizes n | 4 | 5 | 6 | 7 | 8 | 9 | 10 |
|---|---|---|---|---|---|---|---|
| MAPE | 453.66% | 7.36% | 7.29% | 8.76% | 10.14% | 11.21% | 12.45% |

**Table 2. MAPE for different dynamic grey prediction models.**

| Dynamic grey Prediction model | GM | DNGM | DNMGM (n = 5) | DNMGM (n = 6) |
|---|---|---|---|---|
| MAPE | 17.35% | 13.26% | 7.36% | 7.29% |

DNMGM(1,1) model in transportation forecasting, surpassing single static grey prediction models.

A key characteristic of resilient cities is their ability to dynamically adapt to change. The DNMGM(1,1) model has the ability to update historical information to reflect new data, which means that forecasts are not static but can be adjusted based on the most recent data. In urban planning and disaster management, it is not limited to a single dimension of construction safety, but as an important tool for urban resilience construction, it can assist decision-making across fields and dimensions. By updating historical information in real-time to reflect the latest data changes, the DNMGM(1,1) model gives cities the ability to dynamically adapt to challenges, enabling resource allocation and risk response strategies to be flexibly adjusted according to the actual situation, which is of immeasurable value for improving cities' ability to cope with emergencies and ensuring residents' safety and well-being.

## 4.2 The predictive ability of the dynamic grey model under various data conditions

When comparing previous models with the newly proposed DNMGM model, we find that these models show significant differences and similarities in data processing methods,

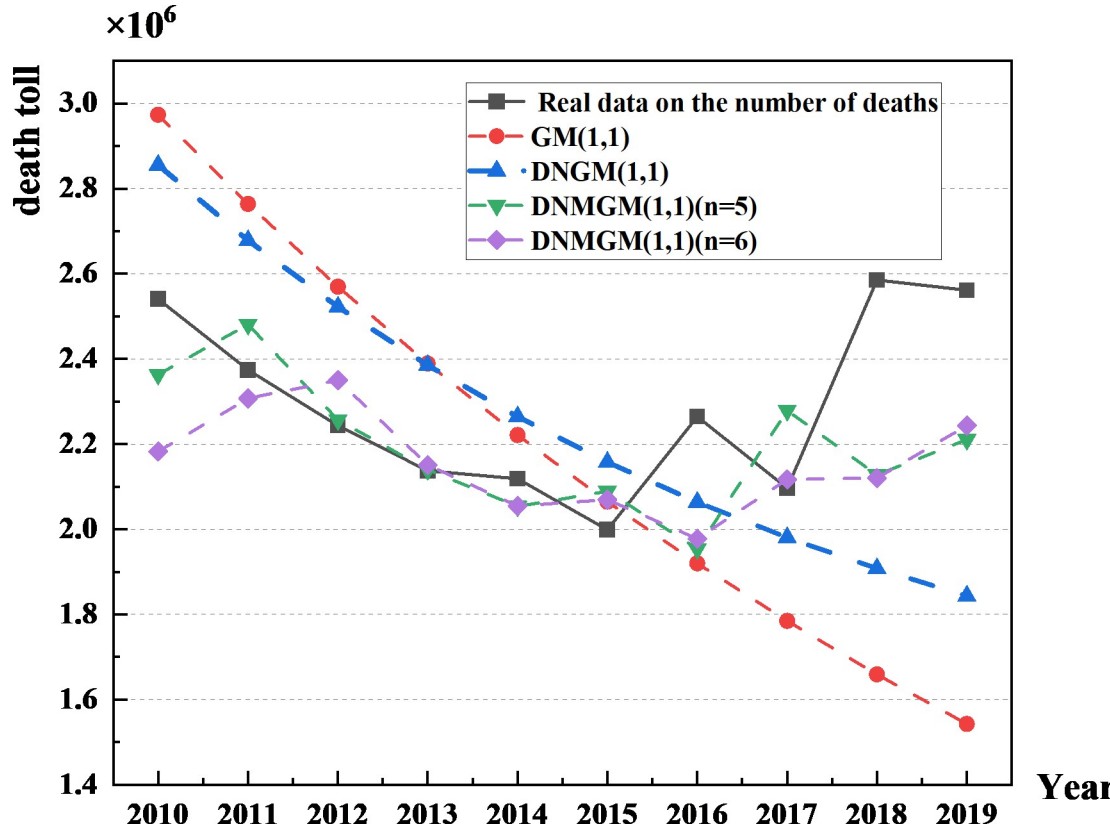

**Fig 8. Comparison of the real and predicted values of traffic accident fatalities in China obtained with the four models.**

prediction frameworks, and prediction accuracy. As a basic grey system forecasting tool, GM is famous for its efficient use of small and incomplete information, but limited by its static characteristics, it is difficult to reflect the new data trend in real-time. DNGM introduces non-homogeneous terms based on GM to better accommodate nonlinear changes in the data, but is again limited by the static framework. MGM (Dynamic Multivariable Grey Model) is an important turning point. It makes the model framework dynamic, allowing the model to integrate new data points in real-time and dynamically adjust internal parameters to track new trends or fluctuations in the data effectively. This dynamic nature greatly improves the adaptability and prediction accuracy of the model when dealing with dynamically changing data. The primary strength of the dynamic grey model lies in its capacity to integrate and promptly respond to newly available data. When new information is or observed data are available, the dynamic grey model can adjust its internal parameters to reflect the current data state and potential trends. This dynamic adjustment mechanism ensures that the model can maintain high prediction accuracy in the face of rapid changes in data.

Furthermore, this adaptation of the dynamic grey model makes it particularly valuable in many domain applications, especially in scenarios in which the data environment is rapidly changing or subject to various external disturbances. For example, in predicting the fatalities in construction and transportation fields in this paper, the volatility and uncertainty of the data are very high, and the traditional static models may fail or produce misleading predictions. Due to its dynamic characteristics, the dynamic grey model can capture these changes more accurately and in a timely manner, thus providing more reliable information for decision-makers.

DNMGM model, as the core innovation of this study, integrates the nonlinear processing capability of DNGM and the dynamic characteristics of MGM to build an efficient prediction model that can not only capture the nonlinear characteristics of data but also respond to data changes in real-time. This combination significantly improves DNMGM's accuracy and computational efficiency compared with other models, especially when dealing with complex and variable construction death data, and its superiority has been fully verified. In terms of accuracy, the DNMGM (1,1) model can effectively adapt to the characteristics of large fluctuations, seasonality, and strong trend of data in the construction field through the nonlinear transformation and dynamic adjustment mechanism so that the prediction accuracy is often better than some simple linear regression models or machine learning algorithms that do not fully consider the dynamic characteristics of data. For example, the performance of basic ANN (Artificial Neural Network) before complex tuning. However, compared to more advanced machine learning algorithms such as R.F. (Random Forest), GBT (Gradient Lift Tree), or deep learning models such as LSTM (Long Short Term Memory Network), the accuracy of DNMGM (1,1) can vary depending on the characteristics of the data. When the data amount is large enough and the feature relationship is complex, these algorithms may show higher prediction accuracy through more complex nonlinear mapping and stronger learning ability. In terms of computing efficiency, DNMGM (1,1) model usually has a faster computing speed and lower resource consumption due to its concise model structure and fewer parameter Settings, which is suitable for application scenarios with high real-time requirements. In contrast, although machine learning algorithms can provide higher prediction accuracy, their computational complexity is often higher, requiring more computational resources and time for model training and prediction, which may be a limiting factor in some situations where real-time performance is strictly required.

In general, G.M. and DNGM models demonstrate the advantages of grey system theory in dealing with uncertainty and small sample data, but are limited by the static framework; The MGM model overcomes this limitation by introducing dynamics. DNMGM model further

combines the advantages of nonlinear and dynamic, and provides a new perspective and tool for the prediction research in the field of building construction safety. The differences and similarities of these models not only reflect the development of grey system theory, but also provide valuable reference and enlightenment for future related research.

## 5 Conclusion

In the face of rapid urbanization, the safety of buildings and transportation systems has become a main issue in urban resilience planning. Resilient city construction requires high degrees of flexibility and adaptability in predicting and responding to future challenges, and traditional static grey models such as G.M. (1,1) have difficulties in dealing with rapidly changing and highly volatile data. The proposed DNMGM(1,1) model, which can effectively overcome the limitations of static models, such as historical data dependence and weak adaptability, combines static optimization with dynamic capabilities to respond to new data and provide high prediction accuracy; thus, its flexibility and applicability are greatly enhanced compared to those of traditional grey models. The specific conclusions are as follows:

1. To precisely forecast the fatality count in construction accidents, we examined five static grey prediction models. The findings indicate that, among these static models, the DNGM (1,1) model stands out as the optimal choice. Notably, the IDGM(1,1) model and the DDGM(1,1) model prove unsuitable for predicting construction accidents in the Chinese context. The MAPE for G.M. (1,1) model and the DGM(1,1) model in forecasting construction accident fatalities in China are recorded at 15.03% and 15.01%, respectively. The DNGM(1,1) model shows superior performance in terms of prediction accuracy, with a MAPE of 13.73%.

2. To compensate for the limitations of static models in responding to long-term data forecasting, we introduce a new dynamic model, DNMGM, with optimal results observed at a data size of 6. Compared with the traditional model, this model integrates and respond to new data points in real-time, which not only accounts for changes in the time series data but also combines the basic principles of grey forecasting and advanced data processing methods to effectively capture dynamic data trends. Based on model comparisons, the DNMGM(1,1) model outperforms others significantly in terms of prediction accuracy, with an average relative error of 9.26%.

3. In addition to its successful application in the construction domain, the DNMGM(1,1) model demonstrates exceptional performance in diverse areas, notably in predicting traffic accident fatalities. At dataset sizes 5 or 6, DNMGM(1,1) exhibits MAPEs of 7.36% and 7.29%, respectively, significantly lower than G.M. model (17.35%) and DNMGM model (13.26%). These outcomes underscore the extensive applicability and robustness of the DNMGM(1,1) model across various domains.

   However, the predictive effect of the DNMGM(1,1) model may be impacted by abnormal data fluctuations resulting from natural disasters, major incidents, or policy changes. To enhance the model's robustness and adaptability, specific strategies should be considered including, but not limited to, incorporating outlier detection and handling mechanisms and exploring hybrid approaches by combining the DNMGM(1,1) model with other prediction methods or algorithms. Future research should delve deeper into the design and implementation of such combination methods, aiming to make the DNMGM(1,1) model more resilient and applicable across various domains.

## Author Contributions

**Conceptualization:** Jian Liu.

**Data curation:** Ye He, Bin Lyu.

**Formal analysis:** Ye He.

**Funding acquisition:** Rui Feng.

**Investigation:** Ye He, Bin Lyu.

**Methodology:** Jian Liu.

**Project administration:** Jian Liu.

**Resources:** Jian Liu.

**Supervision:** Jian Liu.

**Visualization:** Rui Feng.

**Writing – original draft:** Ye He, Rui Feng.

**Writing – review & editing:** Jian Liu, Ye He, Rui Feng, Bin Lyu.

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
