## [Decision Letter · Decision Letter 0]

9 Jul 2024

PONE-D-24-16340Building Safer and More Resilient Cities in China: A Novel Approach Using a Dynamic Nonhomogeneous Gray Model for Data-Driven Decision-MakingPLOS ONE

Dear Dr. FENG,

Thank you for submitting your manuscript to PLOS ONE. After careful consideration, we feel that it has merit but does not fully meet PLOS ONE’s publication criteria as it currently stands. Therefore, we invite you to submit a revised version of the manuscript that addresses the points raised during the review process.

We look forward to receiving your revised manuscript.

Kind regards,

Qing-Chang Lu

Academic Editor

PLOS ONE

Journal Requirements:

"The authors gratefully acknowledge the support by the National Science Foundation of China (Grant No. 52004139) and the National Key R&D Program of China (No.2017YFC0804901), and the Fundamental Research Funds for the Central Universities (No. FRF-TP-22-120A1)."

"The authors received no specific funding for this work."

5. We note that your Data Availability Statement is currently as follows: [All relevant data are within the manuscript and its Supporting Information files.]

Additional Editor Comments:

The manuscript has some values but should be revised based on the reviewers' comments.

Reviewers' comments:

Reviewer's Responses to Questions

**Comments to the Author**

1. Is the manuscript technically sound, and do the data support the conclusions?

Reviewer #1: Yes

Reviewer #2: Partly

2. Has the statistical analysis been performed appropriately and rigorously? 

Reviewer #1: Yes

Reviewer #2: N/A

3. Have the authors made all data underlying the findings in their manuscript fully available?

Reviewer #1: No

Reviewer #2: Yes

4. Is the manuscript presented in an intelligible fashion and written in standard English?

Reviewer #1: Yes

Reviewer #2: Yes

5. Review Comments to the Author

Reviewer #1: This study provides an interesting topic with a concise overview of the research on building safety within the context of urban resilience in China. I think the main point of this research wants to convey the importance of urban resilience in the face of rapid urbanization and the necessity of developing cities that can foresee risks, reduce disaster impacts and recover swiftly from crises. The proposed dynamic nonhomogeneous gray model (DNMGM(1,1)) is clearly presented as the core methodology. Overall, the manuscript is well written, and the implementation and the experiment are an interesting contribution, but there is little original method, and research problem and significance are ambiguous. I have the following detailed concerns:

Abstract: The abstract presents a compelling case for the use of the DNMGM (1,1) model in enhancing urban resilience through better prediction and planning in building safety. While the abstract is largely comprehensive, it could briefly mention any limitations of the study or areas for future research. How the findings can be implemented by urban planners or policymakers might further enhance the practical relevance of the study?

1. Introduction:

This introduction effectively highlights the importance of the construction industry, the significant economic contribution of this sector and the pressing concerns related to construction safety. It provides a well-rounded context by discussing the economic impact, employment generation, and the societal challenges associated with construction safety incidents.

Areas for improvement: The introduction is dense with information and could benefit from clearer, more concise language in some parts. Simplifying the explanation of complex models and their limitations could enhance readability. While the introduction references several studies, integrating these references more seamlessly into the narrative could improve the flow. For instance, directly linking specific studies to the claims made about model performance can strengthen the argument. Explicitly stating the research gap (i.e., the need for a more accurate model for predicting construction fatalities) and the objective of the paper (introducing the DNMGM(1,1) model) earlier in the introduction could provide clearer direction. Briefly mentioning how the new model could be practically implemented in the construction industry and its potential real-world impact would add depth and relevance. Some transitions between paragraphs could be smoother. For example, moving from the general importance of construction safety to specific predictive models feels abrupt. A clearer structure with subheadings might help in organizing the content more logically. By enhancing clarity, conciseness, and integrating practical implications, the introduction can be further strengthened.

2. Approximate inhomogeneous dynamic GM(1,1) modeling process:

While this section is informative, some parts are dense and could be more concise. Simplifying the language and breaking down complex ideas into shorter sentences could improve readability. The transitions between discussing static models, dynamic models, and the specifics of the DNMGM(1,1) model could be smoother. Clearer subheadings might help guide the reader through the different sections. The mention of Fig. 1 is helpful, but a brief description or summary of what the figure illustrates would aid in understanding.

The theoretical justification for the DNMGM(1,1) model is strong, a brief discussion on how this model could be practically implemented in the construction industry would add value. For example, mentioning specific steps or tools for integrating this model into existing safety protocols could enhance its practical relevance. Providing hypothetical or actual examples of how the DNMGM(1,1) model has improved prediction accuracy in comparison to other models would make the argument more compelling.

3. Forecasting of the death risk in China:

This section provides a good overview of the historical trends in construction accident fatalities in China and the rationale for using the gray prediction model. Redundant phrases should be eliminated. Here, authors briefly mentioned various indicators but does not explain why fatalities are the primary focus. A brief justification would be helpful. More detailed description of what Fig. 2 shows would help readers who cannot see the figure immediately. Moreover, providing more context about the data source and type would strengthen the credibility of the data.

4. Static prediction results and analysis:

Some sentences are lengthy and complex, which can affect readability. The explanation of what Fig. 3 and Fig. 4 depict could be more detailed. This would help readers understand the visual data better. Providing a bit more context about why certain models performed better or worse could enhance the reader's understanding of the underlying reasons for these differences. The transition between discussing the performance of the models and the final conclusion could be smoother.

5. Dynamic prediction results and analysis:

Ensure that all technical terms and abbreviations, such as MAPE and DNGM(1,1), are clearly defined when first introduced. Include direct references to the figures in the text to guide the reader (e.g., "As shown in Fig. 5"). Provide a brief explanation of why MAPE is used as the evaluation metric and how it is calculated. Expand the description of each model (GM, DNGM, MGM, DNMGM) to provide more context on their differences and similarities.

By incorporating these revisions, the section will be more informative and easier for readers to understand the significance and implications of the dynamic adaptive capability and generalizability of the DNMGM(1,1) model.

Ensure consistent use of model names (for e.g. Gray/ Grey) and terms throughout the text. Include direct references to figures and tables within the text to guide the reader (e.g., "As shown in Table 1"). Overall, improve the structure and flow of the sections to enhance readability and logical coherence, ensuring smooth transitions between paragraphs and sections. Correct minor typographical errors (e.g., "is or observed data" should be "or observed data is" and Instead of he "The").

Reviewer #2: 1.The use of the dynamic nonhomogeneous gray model (DNMGM(1,1)) represents a significant advancement in predictive modeling for urban resilience. The ability of this model to integrate new data in real-time enhances its applicability in rapidly changing urban environments.

2.The study effectively validates the DNMGM(1,1) model using real-world data on construction and traffic accident fatalities. The reported average relative errors demonstrate the model's superior accuracy compared to traditional gray models, reinforcing its potential for practical implementation.

3.What specific parameters were used in the DNMGM(1,1) model, and how were they determined? Could the choice of these parameters affect the model’s predictive accuracy?

4.What were the primary sources of the data used for model validation? How was data quality ensured, and were there any significant challenges in obtaining reliable data for this study?

5.The research underscores the versatility of the DNMGM(1,1) model by demonstrating its efficacy in different domains, such as construction and traffic safety. This highlights the model's robustness and potential for widespread use in various aspects of urban planning and risk management.

6.How does the DNMGM(1,1) model compare with other advanced predictive models, such as machine learning algorithms, in terms of accuracy and computational efficiency? Are there scenarios where other models might be more suitable?

7.Given the model’s sensitivity to abnormal data fluctuations, what specific strategies or combination methods are being considered to enhance its robustness? How might these strategies be implemented in future research?

8. It is suggested to add articles entitled “Baxhuku et al. New Law Enforcement Impact on the Prevention of Road Accidents in Kosovo”, “Balal et al. Forecasting Solar Power Generation Utilizing Machine Learning Models in Lubbock” and “Al-Abayechi & Al-Khafaji. Forecasting the Impact of the Environmental and Energy Factor to Improve Urban Sustainability by Using (SEM)” to the literature review.

9.The conclusion suggests future research should focus on combining the DNMGM(1,1) model with other predictive methods to address its limitations in handling abnormal data fluctuations. This proactive approach is commendable and necessary for enhancing the model's reliability and applicability.

10.The study presents its findings in a clear and structured manner, making it easy to follow the progression from model development to validation and application. This clarity is essential for understanding the model's capabilities and limitations.

11.What are the practical implications of implementing the DNMGM(1,1) model in urban planning and disaster management? Are there any case studies or pilot projects that demonstrate its real-world applicability and benefits?

6. PLOS authors have the option to publish the peer review history of their article (what does this mean?). If published, this will include your full peer review and any attached files.

Reviewer #1: **Yes: **Rahisha Thottolil

Reviewer #2: No

---

## [Author Response · Author response to Decision Letter 0]

27 Aug 2024

Dear Editor,

Thank you for providing the reviews for our paper entitled " Building Safer and More Resilient Cities in China: A Novel Approach Using a Dynamic Nonhomogeneous Gray Model for Data-Driven Decision-Making."

We deeply appreciate the time and attention the reviewers have devoted to our manuscript. Their valuable comments have significantly contributed to the refinement of our paper and provided essential guidance that has shaped our research. We have thoroughly revised our manuscript based on their suggestions and your specific editorial advice. We have carefully considered each comment and made detailed corrections, which are highlighted in blue in the revised manuscript. Specifically, we have made the following changes:

Page 2, lines 18-21; Page 5, lines 14-25; Page 6, lines 12-13; Page 7, lines 21-28; 

Page 8, lines 1-3; Page 10, lines 10-12; Page 12, lines 10-13; Page 12, lines 15-18;

Page 12, lines 23-27; Page 13, lines 1-21; Page 14, lines 10-18; Page 15, lines 5-14;

Page 15, lines 18-27; Page 17, lines 3-11; Page 17, lines 15-28; Page 18, lines 1-12;

Page 19, lines 13-23; Page 25, lines 8-17; Page 26, lines 3-15; Page 27, lines 6-28;

Page 28, lines 1-16; Page 30, lines 4-11; Details can be viewed in our separate response comments.

We hope these revisions meet your requirements and that our manuscript fully resonates with the journal's focus on building safety. Thank you once again for the opportunity to enhance our manuscript. We are grateful for the editorial guidance and hope our revised submission will be favorable. Please do not hesitate to contact us should you need further modifications or have additional feedback.

Best Regards,

Sincerely,

Rui Feng.

---

## [Decision Letter · Decision Letter 1]

3 Sep 2024

Building Safer and More Resilient Cities in China: A Novel Approach Using a Dynamic Nonhomogeneous Gray Model for Data-Driven Decision-Making

PONE-D-24-16340R1

Dear Dr. FENG,

We’re pleased to inform you that your manuscript has been judged scientifically suitable for publication and will be formally accepted for publication once it meets all outstanding technical requirements.

Kind regards,

Qing-Chang Lu

Academic Editor

PLOS ONE

Additional Editor Comments (optional):

Reviewers' comments:

Reviewer's Responses to Questions

**Comments to the Author**

1. If the authors have adequately addressed your comments raised in a previous round of review and you feel that this manuscript is now acceptable for publication, you may indicate that here to bypass the “Comments to the Author” section, enter your conflict of interest statement in the “Confidential to Editor” section, and submit your "Accept" recommendation.

Reviewer #2: All comments have been addressed

2. Is the manuscript technically sound, and do the data support the conclusions?

Reviewer #2: Yes

3. Has the statistical analysis been performed appropriately and rigorously? 

Reviewer #2: Yes

4. Have the authors made all data underlying the findings in their manuscript fully available?

Reviewer #2: Yes

5. Is the manuscript presented in an intelligible fashion and written in standard English?

Reviewer #2: Yes

6. Review Comments to the Author

Reviewer #2: Review Comments to the Author: Acceted for publication this version. Review Comments to the Author: Acceted for publication this version.

7. PLOS authors have the option to publish the peer review history of their article (what does this mean?). If published, this will include your full peer review and any attached files.

Reviewer #2: No

---

## [Editor Report · Acceptance letter]

27 Sep 2024

PONE-D-24-16340R1 

PLOS ONE

Dear Dr. FENG, 

I'm pleased to inform you that your manuscript has been deemed suitable for publication in PLOS ONE. Congratulations! Your manuscript is now being handed over to our production team.

Kind regards, 

on behalf of

Dr. Qing-Chang Lu 

Academic Editor

PLOS ONE